# Sharpness-Aware Minimization Activates the Interactive Teaching's Understanding and Optimization

**Mingwei Xu** [*][1], **Xiaofeng Cao** [*][†][1], **Ivor W. Tsang** [2,3]

[1] School of Artificial Intelligence, Jilin University, China
[2] CFAR and IHPC, Agency for Science, Technology and Research (A*STAR), Singapore
[3] College of Computing and Data Science, Nanyang Technological University, Singapore
`xumw23@mails.jlu.edu.cn`, `xiaofengcao@jlu.edu.cn`, `ivor_tsang@cfar.a-star.edu.sg`

## Abstract

Teaching is a potentially effective approach for understanding interactions among multiple intelligences. Previous explorations have convincingly shown that teaching presents additional opportunities for observation and demonstration within the learning model, such as data distillation and selection. However, the underlying optimization principles and convergence of interactive teaching lack theoretical analysis, and in this regard co-teaching serves as a notable prototype. In this paper, we discuss its role as a reduction of the larger loss landscape derived from Sharpness-Aware Minimization (SAM). Then, we classify it as an iterative parameter estimation process using Expectation-Maximization. The convergence of this typical interactive teaching is achieved by continuously optimizing a variational lower bound on the log marginal likelihood. This lower bound represents the expected value of the log posterior distribution of the latent variables under a scaled, factorized variational distribution. To further enhance interactive teaching's performance, we incorporate SAM's strong generalization information into interactive teaching, referred as Sharpness Reduction Interactive Teaching (SRIT). This integration can be viewed as a novel sequential optimization process. Finally, we validate the performance of our approach through multiple experiments.

## 1 Introduction

**Backgrounds.** Teaching recognized as a pervasive mechanism for disseminating knowledge within human society, has found extensive application in contemporary deep learning methodologies. It serves as a cornerstone for various techniques such as knowledge distillation [17, 32, 15], data distillation [43], model compression [34, 7], and machine teaching, facilitating optimal training control [46, 47]. Recent investigations into pedagogy have illuminated the integration of large language models and multi-agent systems into educational frameworks. This novel approach emphasizes the significance of understanding interactions among multiple clients and agents. Multi-agent systems, anchored in language models, orchestrate the accomplishment of intricate tasks by assigning specific roles and prescribing behavioral norms to individual agents [18, 37]. Furthermore, they exhibit the capacity to transcend individual intelligence barriers through collaborative endeavors [24, 6], competitive dynamics, deliberative discourses [26, 12, 5], and other strategic modalities.

*Question: However, the optimization and generalization mechanisms concerning interactive teaching strategies remain at an incipient stage of exploration [38]. Consequently, interactive teaching-based*

---

[*]Equal contribution.
[†]Corresponding author.

38th Conference on Neural Information Processing Systems (NeurIPS 2024).

*strategies hold promise in furnishing substantial inductive biases for further advancement in AI agent research. Especially for bidirectional interactive teaching, which lacks sufficient attention and theoretical exploration.*

In the interactive teaching paradigm, the learning algorithm represented primarily by co-teaching [16, 44] has demonstrated notable efficacy in achieving successful learning outcomes in the context of noisy data. This pedagogical approach can be delineated as an interactive teaching prototype wherein two networks with identical architectures serve as peer entities, engaging in interactions aimed at selecting samples with minimal loss values for parameter refinement. Despite the practical efficacy exhibited by interactive teaching methodologies like co-teaching, a notable lacuna persists in terms of theoretical comprehension and requisite convergence analyses. In light of this, we consider co-teaching as an incipient prototype for interactive learning, warranting further scholarly exploration. A more profound understanding of this interactive teaching paradigm holds the potential to yield significant insights into the dynamics governing interactions among intelligent agents [45]. For instance, amid projections of diminishing data availability for training large-scale language models in the foreseeable future and the attendant challenges posed by synthesized data, ongoing research endeavors are directed towards facilitating collaborative engagements between disparate AI models to collectively generate data of higher fidelity and reliability [35, 14].

The exploration of the loss landscape is paramount in elucidating the dynamics of interactive teaching, driving advancements in model generalization. Dinh et al. [10] introduce an important view that favors flat minima over sharp minima in terms of generalization. However, applying this proposition directly is hindered by overlooking the loss landscape geometric properties inherent in commonly used deep architectures. Foret et al. [13] introduce a pioneering methodology termed Sharpness-Aware Minimization (SAM), which entails delineating a perturbation neighborhood in parameter space, identifying the perturbation point that maximizes the loss value, and subsequently optimizing it via gradient descent. This formulation lends itself to a min-max optimization problem, effectively solvable through gradient descent techniques. The conception of SAM inspires contemplation on the interaction and update mechanisms of two networks within the context of interactive, within the framework of the loss landscape. While SAM operates on a single network, it necessitates the computation of two separate gradients at distinct locations. In this case, the initial step involves identifying the sharpest points within a specified range of data proportions within the loss landscape. While diverging from SAM's approach, interactive teaching omits the utilization of these sharp points and instead updates parameters based on the minima of the remaining loss points.

**Our contributions.**   In this paper, we initially ascertain that interactive teaching effectively diminishes the loss landscape by strategically discarding a specific subset of high-loss data points. This process serves to optimize the training procedure, particularly beneficial in scenarios characterized by noisy data, ultimately resulting in a transition to a low-loss landscape during each iterative interaction. Such an approach may be construed as a prior induction of bias, strategically acknowledging the presence of noise within the training dataset. Subsequently, we advance the conceptualization of interactive teaching as an EM-iterative parameter estimation technique, drawing upon seminal work by Dempster et al. [9]. The method's convergence is predicated upon the continual refinement of the lower bound of maximum likelihood estimation. This refinement targets the expectation of the noise posterior distribution pertaining to latent variables, enacted via a relaxation of inequalities. Moreover, to mitigate the inherent challenge of local optima within the interactive framework, we add a level of sharpness knowledge exchange that includes gradient information, which we refer to as **Sharpness Reduction Interactive Teaching (SRIT)**. This amalgamation delineates a novel dual-level sequential optimization paradigm. Finally, empirical validation of our proposed methodology is undertaken across diverse datasets, serving to substantiate its efficacy in augmenting model generalization capabilities. In summation, this manuscript underscores the following key contributions:

- From a perspective centered on loss sharpness, interactive teaching methodologies, such as co-teaching, serve to facilitate parameter adjustments directed at alleviating elevated loss values within the optimization landscape. This mechanism exhibits parallels with SAM optimization, notably in its emphasis on reducing sharpness.

- Our analysis establishes that interactive teaching can be delineated as a probabilistic model, with the incorporation of noisy data as latent variables shedding light on its operational intricacies. This elucidation presents a robust framework for the optimization of interactive teaching methodologies.

- Theoretically, our research confirms that the interactive teaching paradigm and the EM algorithm share certain underlying principles. Combined with the SAM method, it can effectively alleviate the issue of local optima. Therefore, this integration promotes better convergence towards reducing global sharpness in the optimization of loss landscape.

## 2 Related Work

**Interactive teaching** In teaching community, the use of two networks for interactive teaching has gained prominence. Blum and Mitchell [4] divide examples into two views and trained separate algorithms on each, using their predictions to expand the other's training set. To address noisy labels, Malach and Shalev-Shwartz [29] propose a meta-algorithm with two identical predictors that update parameters based on prediction disagreements. Jiang et al. [19] introduce MentorNet, a neural network that guides a deep network (StudentNet) to focus on likely correct samples, reducing overfitting to corrupted labels. Co-teaching [16] employs two networks to combat noisy labels, with each network teaching its peer using small-loss instances. Variants like co-teaching+ [44], JoCoR [36], and CNLCU [40] have emerged. Co-teaching+ selects data with inconsistent predictions, JoCoR promotes prediction consistency and applies constraints, and CNLCU uses interval estimation to account for loss uncertainty. Based on these investigations, this paper adopts co-teaching as the prototype for interactive teaching research. In contrast to the exchange of loss information in a single interaction, our proposed algorithm introduces an additional level of sharpness knowledge exchange that includes gradient information. This can be viewed as a form of dual-level interactive learning.

**Loss landscape** Several studies have explored the relationship between loss landscape flatness and optimization. Li et al. [25] investigate how network structures impact the loss landscape, finding that shortcut connections in ResNet lead to convergence towards better minima, while deeper models have sharper landscapes, and wider models tend to be flatter with better performance. Yao et al. [42] suggest that increasing batch size increases the spectrum of the Hessian matrix, resulting in convergence towards sharper solutions and higher error rates for deeper local minima. Baldassi et al. [2] demonstrate that the error loss function exhibits few extremely wide flat minima and propose entropy-driven algorithms for searching these regions. Bisla et al. [3] derive an optimization algorithm using low-pass filters to actively search for flat regions in the deep learning optimization landscape, similar to SGD.

**Sharpness-Aware Minimization (SAM)** SAM aims to improve the generalization performance of deep neural networks by seeking minima with flatter loss landscapes. Researchers have investigated various aspects related to the weight perturbation radius. Adaptive SAM [22] demonstrates that fixed-radius sharpness is sensitive to parameter rescaling, therefore incorporating scale-invariant adaptive sharpness. Surrogate Gap Minimization [50] defines an easily computable surrogate gap, which is equivalent to the dominant eigenvalue of the Hessian matrix. Du et al. [11] propose Efficient SAM, which incorporates two training strategies: Stochastic Weight Perturbation and Sharpness-Sensitive Data Selection. To alleviate high complexity, Liu et al. [27] propose a novel algorithm called Look-SAM that only periodically calculates the inner gradient ascent. Jiang et al. [20] design an adaptive policy based on the geometric structure of the loss function to enable random or periodic switching between SAM updates and ERM updates. Sparse SAM [31] accelerates training by introducing sparse perturbations through a binary mask. Other perspectives on SAM include Andriushchenko and Flammarion [1], who suggest that a smaller number of data points within each batch can result in better implicit bias of gradient descent for commonly used neural network architectures, and Zhang et al. [48], who propose first-order flatness to bound the maximal eigenvalue of the Hessian at local minima. Additionally, Dai et al. [8] point out that normalization in SAM helps stabilize the algorithm and makes it less sensitive to the choice of the hyperparameter $\rho$.

## 3 Preliminaries

**Non-Convex Optimization for Loss Function** For a training dataset $\mathbb{D} = \{(x_i, y_i)\}_{i=1}^{N}$, where $x_i \in \mathcal{X}$ represents the input and $y_i \in \mathcal{Y}$ represents the outputs or targets. We use $f_\theta(x)$ to denote a commonly used neural network with $\theta$ representing the weight parameters of the network. $\ell : \theta \times \mathcal{X} \times \mathcal{Y} \to \mathbb{R}_+$ represents a per-data-point loss function, the training of neural network is

typically a non-convex optimization problem, the *empirical risk minimization* is shown as Equation 1:

$$\hat{\theta} = \arg\min_{\theta} \ \mathcal{L}_{\mathbb{D}}\left(f_{\theta}\right) \quad \text{where} \quad \mathcal{L}_{\mathbb{D}}\left(f_{\theta}\right) = \frac{1}{|\mathbb{D}|} \sum_{(x_i, y_i) \in \mathbb{D}} \ell\left(f_{\theta}\left(x_i\right), y_i\right), \tag{1}$$

we make the assumption that the function $\mathcal{L}_{\mathbb{D}}\left(f_{\theta}\right)$ is both continuous and differentiable. During each iteration, the optimizers randomly select a mini-batch $\mathbb{B}_t$ from the set $\mathbb{D}$, using a fixed batch size.

**Sharpness-Aware Minimization (SAM)** SAM [13] expects the training process to unfold in a flatter region, resulting in smaller training losses around the neighborhood of the converged parameter $\hat{\theta}$ and improving the model's generalization performance. The sharpness measure term is defined as the maximum change between the loss caused by parameter perturbation within a neighborhood and the previous loss, expressed as: $\max_{\epsilon:\|\epsilon\|_2 \leq \rho} \mathcal{L}_{\mathbb{D}}(f_{\theta+\epsilon}) - \mathcal{L}_{\mathbb{D}}(f_{\theta})$. SAM optimizes a min-max problem as depicted in Equation 2:

$$\hat{\theta} = \arg\min_{\theta} \max_{\epsilon:\|\epsilon\|_2 \leq \rho} \mathcal{L}_{\mathbb{D}}^{SAM}(f_{\theta+\epsilon}) + \lambda\|\theta\|_2^2, \tag{2}$$

where $\rho$ represents a pre-defined constant that limits the radius of the neighborhood, while $\epsilon$ denotes the weight perturbation vector responsible for maximizing the training loss within the neighborhood constrained by $\rho$. $\lambda$ denotes hyperparameter that dominates a $L_2$ regularization term about weights. $\hat{\epsilon}$ is obtained by approximating the first-order Taylor expansion of $\mathcal{L}_{\mathbb{D}}(f_{\theta+\epsilon})$ at $\theta$ and solving it as a classical dual norm problem:

$$\hat{\epsilon} = \arg\max_{\epsilon:\|\epsilon\|_2 < \rho} \mathcal{L}_{\mathbb{D}}(f_{\theta+\epsilon}) \approx \rho \frac{\nabla_{\theta}\mathcal{L}_{\mathbb{D}}\left(f_{\theta}\right)}{\|\nabla_{\theta}\mathcal{L}_{\mathbb{D}}\left(f_{\theta}\right)\|_2^2}, \tag{3}$$

after obtaining $\hat{\epsilon}$, the outer optimization $\min_{\theta}$ in SAM follows the usual gradient descent update for $\theta$: $\theta_{t+1} = \theta_t - \eta g(\theta + \hat{\epsilon})$, where $\eta$ is the learning rate. The difference lies in the computation of $g(\theta + \hat{\epsilon}) = \nabla_{\theta}\mathcal{L}_{\mathbb{D}}(f_{\theta})|_{\theta+\hat{\epsilon}}$, where the gradient is evaluated at $\theta + \hat{\epsilon}$.

**Co-teaching** The architecture of co-teaching [16] consists of two models, $f$ and $g$, along with their corresponding weights $\theta_f$ and $\theta_g$. The algorithm starts by shuffling the training set $\mathcal{D}$, which represents a noisy dataset. In each iteration, the algorithm fetches a mini-batch $\bar{\mathcal{D}}$ from the shuffled dataset. Next, it selects a subset $\bar{\mathcal{D}}_f$ and $\bar{\mathcal{D}}_g$ of $\bar{\mathcal{D}}$ by choosing the instances with the smallest losses based on the models $f$ and $g$ respectively. This selection is done by randomly sampling a fraction $R(T)$ of the instances with the smallest losses:

$$\bar{\mathcal{D}}_f = \arg\min_{\mathcal{D}':|\mathcal{D}'|\geq R(T)|\bar{\mathcal{D}}|} \mathcal{L}(f, \mathcal{D}'); \quad \bar{\mathcal{D}}_g = \arg\min_{\mathcal{D}':|\mathcal{D}'|\geq R(T)|\bar{\mathcal{D}}|} \mathcal{L}(g, \mathcal{D}'), \tag{4}$$

where $R(t)$ is a parameter that determines the proportion of the smallest values to retain. $|\bar{\mathcal{D}}|$ represents the number of samples in the dataset $\bar{\mathcal{D}}$. Then, the algorithm updates the weights $\theta_f$ and $\theta_g$ by applying a gradient descent step using the selected subsets $\bar{\mathcal{D}}_g$ and $\bar{\mathcal{D}}_f$ respectively:

$$\theta_f = \theta_f - \eta\nabla\mathcal{L}(f, \bar{\mathcal{D}}_g); \quad \theta_g = \theta_g - \eta\nabla\mathcal{L}(g, \bar{\mathcal{D}}_f). \tag{5}$$

The algorithm updates the value of $R(T) \in (0, 1]$ based on the current epoch $T$ and a predetermined parameter $T_k$. The update rule is given by $R(T) = 1 - \min\left\{\frac{T}{T_k}\tau, \tau\right\}$.

## 4 Theoretical Analysis and Solution

In this section, we put forward three main points of view:

1. Interactive teaching methods like co-teaching update parameters by reducing high loss values in the landscape. By actively involving two teachers, models in interactive framework learn from each other's strengths through a collaborative filtering mechanism and focus on minimizing loss examples.

2. Our core assumption is that the cleanliness of the data distribution serves as a latent variable, as it remains unknown within the training dataset. Based on this assumption, the interactive teaching process can be effectively exemplified as a unique type of parameter iteration within the EM framework. This perspective provides a probabilistic modeling-based explanation for the iteration and convergence of interactive teaching, such as co-teaching.

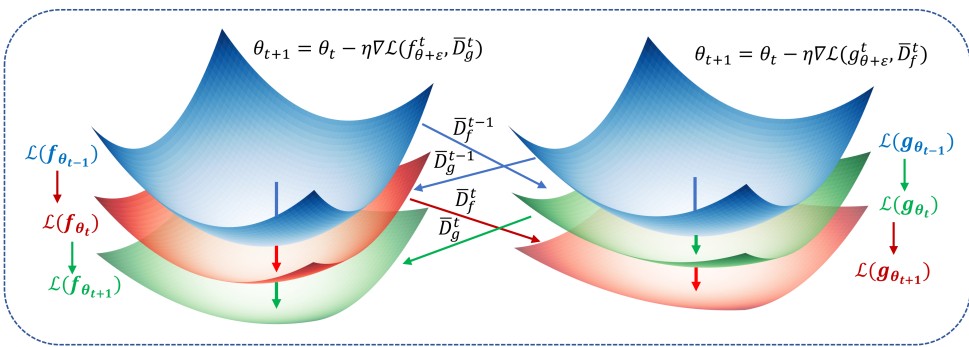

Figure 1: Interactive teaching, Sharpness Reduction Interactive Teaching (SRIT), the plane in the figure represents the loss landscape, which gradually becomes flat during the iterative optimization process due to the receipt of flat gradient information cues from each other.

3. Concerning the local convergence in EM, we observe that a flatter loss landscape facilitates the optimization process in escaping local optima. Under such conditions, we incorporate the SAM to flatten the loss landscape and promote more global convergence within the interactive teaching paradigm. Building upon the exchange of loss information in a single interaction, an additional level of sharpness knowledge exchange containing gradient information has been introduced, which can be regarded as a form of dual-level interactive learning. This interactive teaching process, augmented by SAM, enables the acceptance of flatter high-probability regions from the peer network, thereby enhancing both the predictive performance and generalization capability of the model, as illustrated in Figure 1.

The essence of the SAM method lies in a more refined exploration of the gradient space, implicitly utilizing second-order information about the parameter space in the loss landscape. In experiments, it significantly improves generalization performance but incurs some unavoidable additional computations. The computational power consumption of the current algorithm complexity can be kept within a reasonable range and does not exponentially increase with the scale of data and models. This increase in complexity, compared to computational resources, is considered acceptable.

## 4.1 Analysis of parameter update mechanism

**The critical first step** While both co-teaching and SAM update network weights in two steps per iteration, they differ fundamentally in how they compute the first step. Specifically, in co-teaching, during the selection process of data points with small losses, the network is pretrained and kept fixed. In SAM, the training data source domain remains unchanged, but perturbations are applied to the network weights. Within a set distance $\rho$ neighborhood, SAM seeks the direction of maximum offset $\epsilon$ that induces the greatest change in loss values compared to the original loss, i.e., $\arg\max_{\|\epsilon\|_2 \leq \rho} \mathcal{L}_{\mathbb{D}}(f_{\theta+\epsilon}) - \mathcal{L}_{\mathbb{D}}(f_\theta)$. SAM not only seeks out points within a defined domain where the loss function undergoes the greatest change, exhibiting high curvature characteristics geometrically, but also aims to minimize the value of the loss function in regions of high curvature (sharp regions). The computation $\nabla\mathcal{L}_{\mathbb{D}}(f_\theta)|_{\theta+\epsilon}$, implicitly depends on the Hessian properties of $\mathcal{L}_{\mathbb{D}}(f_\theta)$, as the perturbation value $\hat{\epsilon}$ is determined by the gradient $\nabla\mathcal{L}_{\mathbb{D}}(f_\theta)$.

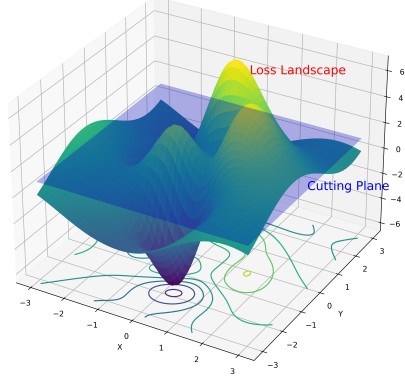

Figure 2: Loss landscape, we can broadly assume that in each iteration, a fixed cutting plane is used to remove the peaks with high loss values.

**Loss landscape perspective** In the non-convex optimization of neural networks, the relationship between the magnitude of the loss value and its gradient is not clearly causal. Let's consider network $f_\theta$ and a set of samples $(x_i, y_i) \in \bar{D}$, and $i = 1, .., N$. The sum of the gradients of the loss function

can be represented by the following Equation 6. The network $g$ undergoes the same process:

$$\nabla\mathcal{L}(f_\theta, \bar{D}) = \sum_{i=1}^{N} \nabla\mathcal{L}_i(f_\theta(x_i), y_i) = \nabla\mathcal{L}(f_\theta(x_1), y_1) + ... + \nabla\mathcal{L}(f_\theta(x_N), y_N), \quad (6)$$

after receiving $K$ data points $\hat{D}_g = \arg\min \mathcal{L}(g_\theta, \bar{D})$ from network $g_\theta$, $(x_j, y_j) \in \hat{D}_g, j = 1, .., K$, we have:

$$\nabla\mathcal{L}(f_\theta, \hat{D}_g) = \nabla\mathcal{L}(f_\theta(x_1), y_1) + ... + \nabla\mathcal{L}(f_\theta(x_j), y_j) + ... + \nabla\mathcal{L}(f_\theta(x_K), y_K), \quad (7)$$

further, according to Equation 6 and 7, we have:

$$\underbrace{\nabla\mathcal{L}(f_\theta, \hat{D}_g)}_{R(t)\cdot\#(\mathcal{L})} = \nabla\mathcal{L}(f_\theta, \bar{D}_g) - \left[\nabla\mathcal{L}(f_\theta(x_{K+1}), y_{K+1}) + ... + \nabla\mathcal{L}(f_\theta(x_N), y_N)\right], \quad (8)$$

$$= \nabla\mathcal{L}(f_\theta, \bar{D}_g) - \underbrace{\nabla\mathcal{L}(f_\theta, \tilde{D}_g)}_{(1-R(t))\cdot\#(\mathcal{L})}, \quad (9)$$

where $\tilde{D}_g = \bar{D}_g \backslash \hat{D}_g = \{(x_{K+1}, y_{K+1}), ..., (x_N, y_N)\}$, the symbol $\#(\mathcal{L})$ represents the quantity of data point loss values. Therefore, for the update at time $t+1$, gradients are computed on examples with low loss values:

$$\theta_{t+1}^f = \theta_t^f - \eta\nabla\mathcal{L}(f_\theta, \hat{D}_g), \quad (10)$$

$$= \theta_t^f - \eta\left(\nabla\mathcal{L}(f_\theta, \bar{D}_g) - \nabla\mathcal{L}(f_\theta, \tilde{D}_g)\right) = \theta_t^f - \eta\nabla\left(\mathcal{L}(f_\theta, \bar{D}_g) - \mathcal{L}(f_\theta, \tilde{D}_g)\right). \quad (11)$$

As illustrated by the gradient decomposition discussed above, it can be observed that co-teaching segments the loss landscape by reducing high loss values and only optimizes parameters on the low loss landscape. This concept is visually depicted in the schematic diagram presented in Figure 2.

## 4.2 EM iterative process in typical interactive teaching

The analysis in Section 4.1 reveals that the interactive teaching utilizes low-loss samples from a peer network as prior knowledge, subsequently refining model parameters. This approach mirrors the iterative parameter estimation of the EM algorithm, commonly employed for probabilistic models with hidden variables. To elucidate the iterative steps and convergence, we adopt the EM framework, grounded in Maximum Likelihood Estimation. In typical interactive teaching, specifically co-teaching, we treat the cleanliness of training data as hidden variables, with $\theta_f$ and $\theta_g$ representing the parameters to be estimated for two structurally identical neural networks. The iterative convergence of the interactive teaching process within the EM framework is then characterized by the following proposition.

**Proposition 4.1.** *Given the training dataset $\mathcal{D} = \{X_i = (x_i, y_i)\}_{i=1}^{N}$, which contains noisy samples, and assuming the samples are independent, we define the hidden variable $Z_c = 1$ to indicate that the c-th sample is a cleaner sample, meaning it has a lower loss value compared to noisy data. $Z = Z_c^f \cup Z_c^g = \{Z_c\}_{c=1}^{K}$ represents the set of hidden variables for all samples, and the corresponding latent distribution is denoted as $q(Z)$. The joint probability of $p(X_i, Z_c|\theta_f, \theta_g)$ is obtained by simultaneously updating neural networks $f$ and $g$ for $X_i$ and $Z_c$. The logarithm likelihood of the observed data $\mathcal{D}$ has the following lower bound $L$:*

$$L(\theta_f, \theta_g, q) \equiv \sum_{i=1}^{N}\sum_{c=1}^{K}\left[q(Z_c)\log\frac{p(X_i, Z_c|\theta_f, \theta_g)}{q(Z_c)}\right], \quad (12)$$

*the EM algorithm approximates the maximization of $\log p(\mathcal{D}|\theta_f, \theta_g)$ by maximizing this lower bound $L(\theta_f, \theta_g, q)$. Specifically, the EM iteration process for the interactive teaching paradigm is as follows:*

**E-step:**

$$Q(\theta_f, \theta_g^{(t)}) = \mathbb{E}_{Z_c^f|\mathcal{D},\theta_f^{(t)},\theta_g}[\log p(Z_c^f, \mathcal{D}|\theta_f, \theta_g^{(t)})], \quad (13)$$

$$Q(\theta_f^{(t)}, \theta_g) = \mathbb{E}_{Z_c^g|\mathcal{D},\theta_f,\theta_g^{(t)}}[\log p(Z_c^g, \mathcal{D}|\theta_f^{(t)}, \theta_g)], \quad (14)$$

*The subscript $Z_c^f | \mathcal{D}, \theta_f^{(t)}, \theta_g$ (or $Z_c^g | \mathcal{D}, \theta_f, \theta_g^{(t)}$) of the expectation represents the corresponding set of low-loss samples selected by network $f$ (or $g$) and is used to update network $g$ (or $f$).*

***M-step:***

$$\theta_f^{(t+1)} = \arg \max_{\theta_f} Q(\theta_f^{(t)}, \theta_g), \tag{15}$$

$$\theta_g^{(t+1)} = \arg \max_{\theta_g} Q(\theta_f, \theta_g^{(t)}). \tag{16}$$

Please refer to Appendix A for the deduction process of the proposition.

*Remark* 4.2. (1) In the E-step, we first calculate the distribution of the hidden variable $p(Z_c^f | \mathcal{D}, \theta_f^{(t)}, \theta_g)$ using the current fixed parameters $\theta_f^{(t)}$ and the dataset $\mathcal{D}$. In the context of interactive teaching, this involves selecting a specific proportion of clean samples to provide to the peer network. We then utilize this posterior distribution to compute the expectation of $\log p(Z_c^f, \mathcal{D} | \theta_f, \theta_g^{(t)})$.

(2) In the M-step, $\theta_f^{(t+1)} = \arg \max_{\theta_f} Q(\theta_f^{(t)}, \theta_g)$ optimizes the parameters of the $f$ network to maximize the logarithm likelihood function on the subset of samples $Z_c^g$ selected by the $g$ network. The network $g$ follows the same process. And as we know, maximum likelihood estimation is equivalent to minimizing the empirical loss.

(3) The key distinction of interactive teaching from the conventional EM algorithm lies in one party receiving data information on the latent variable distribution from the other party. It not only allows for a priori determination of which important information needs to be preserved but also introduces a distribution from the counterpart, thereby increasing randomness and diversity to prevent overfitting. From a probabilistic modeling perspective, one party receives the high-probability region of clean samples from the other party, which effectively enhances the overall performance.

---

**Algorithm 1:** Sharpness Reduction Interactive Teaching (SRIT)

---

**Input:** Initial network parameters $\theta_{f_0}, \theta_{g_0}$, learning rate $\eta$, fixed parameter $\tau$, iteration counts $T_k$ and $T_{\max}$, maximum iteration count $N_{\max}$, pre-defined constant $\rho$.
**Output:** Updated network parameters $\theta_f$ and $\theta_g$.

1 **for** $T = 1$ **to** $T_{\max}$ **do**
2    Shuffle the training set $\mathcal{D}$ (noisy dataset);
3    **for** $N = 1$ **to** $N_{\max}$ **do**
4      Sample a mini-batch $\bar{\mathcal{D}}$ from $\mathcal{D}$;
5      **Dual-level optimization: The first level of loss information exchange.**
6      Compute the loss of network $f$ on $\bar{\mathcal{D}}$ and obtain $\hat{\mathcal{D}}_f$:
       $\hat{\mathcal{D}}_f = \arg \min_{\mathcal{D}':|\mathcal{D}'| \geq R(T)|\bar{\mathcal{D}}|} \mathcal{L}(f, \bar{\mathcal{D}})$;     //sample $R(T) \cdot |\bar{\mathcal{D}}|$ small-loss instances;
7      Compute the loss of network $g$ on $\bar{\mathcal{D}}$ and obtain $\hat{\mathcal{D}}_g$:
       $\hat{\mathcal{D}}_g = \arg \min_{\mathcal{D}':|\mathcal{D}'| \geq R(T)|\bar{\mathcal{D}}|} \mathcal{L}(g, \bar{\mathcal{D}})$     //sample $R(T) \cdot |\bar{\mathcal{D}}|$ small-loss instances;
8      **Dual-level optimization : The second level of sharpness element exchange.**
9      Update $\hat{\epsilon}(\theta_f) = \rho \frac{\nabla_\theta \mathcal{L}_{\hat{\mathcal{D}}_g}(f_\theta)}{\left\| \nabla_\theta \mathcal{L}_{\hat{\mathcal{D}}_g}(f_\theta) \right\|_2^2}$ and $\hat{\epsilon}(\theta_g) = \rho \frac{\nabla_\theta \mathcal{L}_{\hat{\mathcal{D}}_f}(g_\theta)}{\left\| \nabla_\theta \mathcal{L}_{\hat{\mathcal{D}}_f}(g_\theta) \right\|_2^2}$ ;
10      Compute the approximate gradient for network $f$: $\boldsymbol{G}_f = \nabla_{\theta_f} \mathcal{L}(f_\theta, \hat{\mathcal{D}}_g)|_{\theta_f + \hat{\epsilon}(\theta_f)}$;
11      Compute the approximate gradient for network $g$: $\boldsymbol{G}_g = \nabla_{\theta_g} \mathcal{L}(g_\theta, \hat{\mathcal{D}}_f)|_{\theta_g + \hat{\epsilon}(\theta_g)}$;
12      Update the network parameters of $f$ using gradient descent: $\theta_f = \theta_f - \eta \boldsymbol{G}_f$;
13      Update the network parameters of $g$ using gradient descent: $\theta_g = \theta_g - \eta \boldsymbol{G}_g$;
14    **end**
15    Compute $R(T) = 1 - \min \left\{ \frac{T}{T_k}\tau, \tau \right\}$.
16 **end**

---

## 4.3 Sharpness Reduction Interactive Teaching (SRIT)

Since the convergence of the EM algorithm guarantees only local optima, while SAM can flatten the loss landscape and effectively alleviate local optima, favoring global optima, we incorporate SAM into

the interactive teaching process, which referred as Sharpness Reduction Interactive Teaching (SRIT) to enhance performance and generalization. We conclude the proposed method as a novel dual-level sequential optimization process. For the first level, we start by screening the required low-loss dataset $\min_{\hat{\mathcal{D}}} \mathcal{L}(f_\theta, \mathcal{D})$. For the second level, we transfer the loss information to the counterpart model for SAM optimization: $\theta^* = \arg\min_{\hat{\theta}} \max_{\hat{\epsilon}} \mathcal{L}(g_{\theta+\epsilon}, \hat{\mathcal{D}})$. Among the second level, we have two crucial steps: 1).The first step estimates the direction of the change in the network weights $\hat{\epsilon}(\theta_f)$ and $\hat{\epsilon}(\theta_g)$ based on the gradient information of the loss function, and then computes the new approximate gradients $\boldsymbol{G}_f$ and $\boldsymbol{G}_g$,

$$\boldsymbol{G}_f = \nabla_{\theta_f}\mathcal{L}(f_\theta, \hat{\mathcal{D}}_g)|_{\theta_f + \hat{\epsilon}(\theta_f)}, \boldsymbol{G}_g = \nabla_{\theta_g}\mathcal{L}(g_\theta, \hat{\mathcal{D}}_f)|_{\theta_g + \hat{\epsilon}(\theta_g)}, \tag{17}$$

where $\hat{\epsilon}(\theta_f) = \rho \frac{\nabla_\theta \mathcal{L}_{\hat{\mathcal{D}}_g}(f_\theta)}{\left\|\nabla_\theta \mathcal{L}_{\hat{\mathcal{D}}_g}(f_\theta)\right\|_2^2}, \hat{\epsilon}(\theta_g) = \rho \frac{\nabla_\theta \mathcal{L}_{\hat{\mathcal{D}}_f}(g_\theta)}{\left\|\nabla_\theta \mathcal{L}_{\hat{\mathcal{D}}_f}(g_\theta)\right\|_2^2}$. 2). The second step involves updating the parameters of networks $f$ and $g$ based on the estimated gradients. The detailed algorithm procedure is shown in Algorithm 1.

In summary, it involves two levels of interaction: the first level is the exchange of loss information, and the second level is the exchange of sharpness knowledge containing gradient information. The first level filters out data that are harmful to the model, while the second level flattens the optimized loss landscape, making it less prone to local optima, thereby enhancing optimization and generalization performance.

## 5   Experiments

In this section, we will conduct experiments on two core baselines, co-teaching [16] and CNLCU [40]. Co-teaching has served as a foundation for the development of various optimization techniques within this framework, and CNLCU is a cutting-edge research achievement. In the experiment, we use four NVIDIA RTX 6000 GPUs with 24GB of memory each.

Table 1: Test accuracy (%) on five datasets. The best results are highlighted in bold.

| Noise type | Symmetric. | | Pairflip. | | Tridiagonal. | | Instance. | |
|---|---|---|---|---|---|---|---|---|
| Noise ratio | 20% | 40% | 20% | 40% | 20% | 40% | 20% | 40% |
| MNIST | | | | | | | | |
| Co-teaching | 97.50 ±0.06 | 94.96. ±0.07 | 95.49 ±0.11 | 91.54 ±0.15 | 96.61 ±0.06 | 92.76 ±0.09 | 95.90 ±0.05 | 91.23 ±0.18 |
| SRIT | **99.42** **±0.03** | **99.19** **±0.03** | **99.35** **±0.02** | **98.14** **±0.07** | **99.47** **±0.03** | **98.75** **±0.05** | **99.43** **±0.02** | **98.03** **±0.10** |
| CIFAR10 | | | | | | | | |
| Co-teaching | 82.15 ±0.09 | 77.38 ±0.15 | 82.32 ±0.08 | 75.37 ±0.14 | 82.77 ±0.07 | 76.41 ±0.17 | 81.86 ±0.12 | 73.61. ±0.25 |
| SRIT | **85.64** **±0.15** | **79.83** **±0.12** | **85.10** **±0.20** | **76.95** **±0.17** | **85.39** **±0.18** | **78.90** **±0.12** | **84.77** **±0.19** | **74.07** **±0.25** |
| CIFAR100 | | | | | | | | |
| Co-teaching | 50.21 ±0.23 | 42.40 ±0.16 | 48.27 ±0.11 | 34.74 ±0.13 | 50.32 ±0.19 | 38.78 ±0.16 | 49.74 ±0.18 | 38.57 ±0.12 |
| SRIT | **59.66** **±0.16** | **50.57** **±0.21** | **57.16** **±0.10** | **35.82** **±0.16** | **59.07** **±0.15** | **42.27** **±0.22** | **59.66** **±0.16** | **40.36** **±0.18** |
| FMNIST | | | | | | | | |
| Co-teaching | 91.13 ±0.09 | 87.99 ±0.09 | 89.83 ±0.10 | 85.44 ±0.12 | 90.42 ±0.07 | 86.09 ±0.09 | 90.27 ±0.12 | 85.63 ±0.13 |
| SRIT | **92.68** **±0.10** | **88.77** **±0.13** | **92.76** **±0.09** | **89.25** **±0.11** | **91.44** **±0.11** | **89.97** **±0.09** | **91.44** **±0.09** | **86.02** **±0.21** |
| SVHN | | | | | | | | |
| Co-teaching | 91.83 ±0.08 | 88.72 ±0.10 | 91.49 ±0.10 | 85.09 ±0.15 | 92.16 ±0.10 | 87.51 ±0.13 | 91.26 ±0.18 | 86.33 ±0.23 |
| SRIT | **94.95** **±0.05** | **93.06** **±0.05** | **94.34** **±0.08** | **89.37** **±0.20** | **94.66** **±0.05** | **91.56** **±0.12** | **94.45** **±0.08** | **90.50** **±0.10** |

## 5.1 Experimental Settings

**Datasets and type of noise**     Based on previous research [16, 44, 40], we conduct experiments on five widely used datasets to effectively demonstrate the efficacy of the co-teaching algorithm. These datasets include MNIST [23], FMNIST [41], CIFAR10 [21], SVHN [33], and CIFAR100 [21]. In co-teaching, we do not use validation dataset as in research [16]. However, to maintain consistency with CNLCU, we use 90% of the training data and 10% as the validation set in CNLCU. In CNLCU, we conduct experiments on three representative datasets: MNIST, CIFAR10, and CIFAR100. We utilize various types of noise commonly used in multiple studies [28, 44, 39, 49, 40], including symmetric noise, tridiagonal noise, pairflip noise, and instance noise. To facilitate comparison with previous research [40], we set the noise rates in the datasets to 20% and 40% respectively. On the test dataset, we consider the average test accuracy of the last ten epochs as the final test accuracy, accompanied by a 95% confidence interval.

**Models and hyper-parameters**     For all datasets, we utilize a 9-layer CNN architecture [16] with dropout and batch normalization for the classification task. In co-teaching, for all datasets, we use the Adam optimizer with a momentum of 0.9, an initial learning rate of 0.001, and trained for 200 epochs. For $R(T) = 1 - \min\left\{\frac{T}{T_k}\tau, \tau\right\}$, where $T_k$ is set to 10 by default [16]. In SAM related optimization, such as SRIT and SRCNLCU, we use an SGD optimizer with an initial learning rate of 0.1, momentum of 0.9, weight decay of 0.0001, epochs of 200, and set $\rho$ to 0.05 [13]. It has been pointed out by Andriushchenko and Flammarion [1] that the generalization performance of a model is influenced by the number of data points within a batch. Insufficient data quantity in a batch leads to inefficient utilization of GPU accelerators, while excessively large data quantity can result in suboptimal generalization. Therefore, taking reference from [1], we empirically set the batch size to 128 as an optimal choice.

Table 2: Test accuracy (%) on *MNIST,CIFAR10,CIFAR100*. The best results are highlighted in bold.

| Noise type | Symmetric. | | Pairflip. | | Tridiagonal. | | Instance. | |
|---|---|---|---|---|---|---|---|---|
| Noise ratio | 20% | 40% | 20% | 40% | 20% | 40% | 20% | 40% |
| MNIST | | | | | | | | |
| CNLCU-H | 98.70 ±0.06 | 98.24 ±0.06 | 98.44 ±0.19 | 97.37 ±0.32 | 98.89 ±0.15 | 97.92 ±0.05 | 98.74 ±0.16 | 97.42 ±0.39 |
| SRCNLCU-H | **99.16** **±0.02** | **98.81** **±0.05** | **99.01** **±0.03** | **98.38** **±0.20** | **99.04** **±0.03** | **98.42** **±0.05** | **98.88** **±0.03** | **97.84** **±0.04** |
| CIFAR10 | | | | | | | | |
| CNLCU-H | 83.03 ±0.47 | 78.33 ±0.50 | 83.39 ±0.68 | 73.40 ±1.53 | 82.52 ±0.71 | 74.79 ±1.13 | 81.93 ±0.25 | 73.58 ±1.39 |
| SRCNLCU-H | **85.43** **±0.11** | **80.88** **±0.16** | **84.89** **±0.12** | **75.19** **±0.27** | **85.35** **±0.14** | **78.94** **±0.12** | **83.87** **±0.11** | **75.49** **±0.16** |
| CIFAR100 | | | | | | | | |
| CNLCU-H | 46.27 ±0.38 | 42.05 ±0.87 | 43.25 ±0.75 | 30.79 ±0.86 | 45.02 ±1.06 | 35.24 ±0.93 | 45.02 ±1.07 | 36.17 ±1.54 |
| SRCNLCU-H | **55.84** **±0.24** | **44.72** **±0.43** | **53.33** **±0.22** | **33.03** **±0.27** | **54.28** **±0.19** | **38.81** **±0.42** | **54.98** **±0.15** | **37.88** **±0.17** |

## 5.2 Experimental results on SRIT, Co-teaching, SRCNLCU and CNLCU

In this part, our experimental results are saved in Table 1 and 2. In Figure 3, we showcase the test performance on co-teaching and SRIT, more specific details are presented in Appendix B. Under all dataset and noise conditions, SRIT and SRCNLCU with SAM both consistently achieve significantly higher test accuracy compared to using co-teaching and CNLCU alone. This advantage is evident across different types of noise and noise ratios. Additionally, it is apparent from the figures presented in Appendix B that incorporating SAM into the training process demonstrates remarkable generalization capabilities, effectively mitigating overfitting.

**Performance on Different Datasets**     SRIT achieves exceptional performance on the MNIST dataset. For the CIFAR10 dataset, SRIT also performs well. This indicates that even on the more complex CIFAR10 dataset, employing SAM in interactive teaching (co-teaching) can significantly enhance

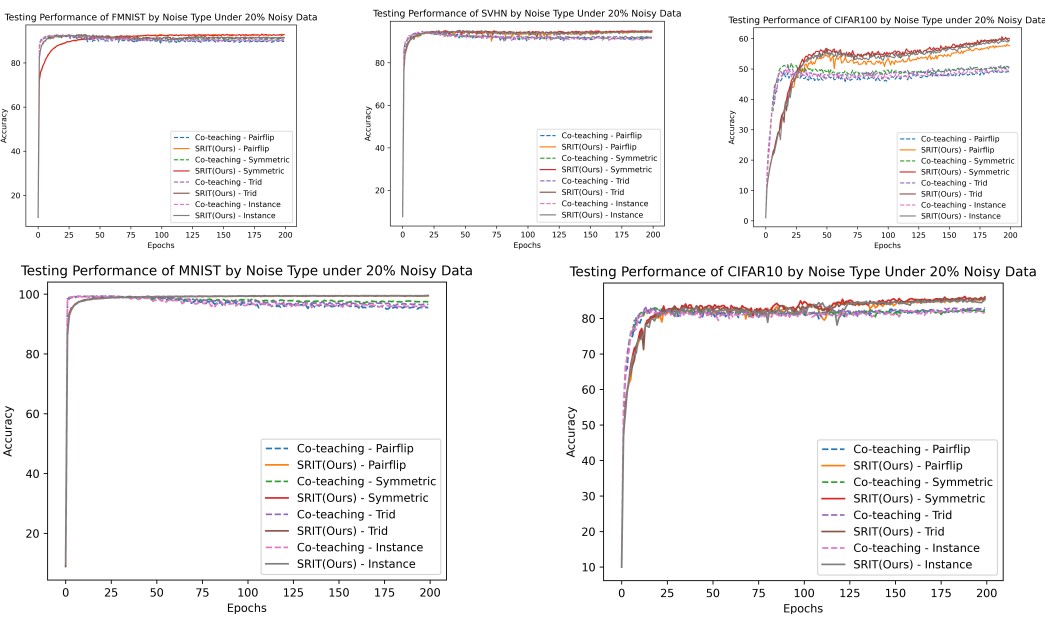

Figure 3: Testings of five datasets by noise type, and the noise ratio is 20%.

the model's generalization ability and accuracy. Similarly, on the FMNIST, SVHN, and CIFAR100 datasets, SRIT outperforms co-teaching in all noise scenarios. In SRCNLCU, nearly all experiments demonstrate a significant advantage. It should be noted that we selected the experimental results of CNLCU-H directly for comparison, without any bias.

From a macro-level perspective, co-teaching and CNLCU with SAM consistently outperformed co-teaching alone across all tested datasets and noise conditions. Whether it is the simple MNIST dataset or the complex CIFAR10 and CIFAR100 datasets, SRIT demonstrate strong robustness and high accuracy. The introduction of the sharpness reduction strategy effectively improves the performance of interactive teaching methods such as co-teaching and CNLCU, particularly when faced with high noise ratios and complex types of noise, resulting in even more significant enhancements.

## 6  Conclusions

In this paper, we first analyze how the low-loss selection of noisy data for interactive teaching reduces high-loss regions. Then, we introduce the EM framework to explore the interactive teaching mechanism, using the co-teaching algorithm as an example, which is a typical algorithm within the interactive teaching paradigm. We demonstrate that the iteration process of the typical interactive teaching algorithm follows the EM algorithm, ensuring its convergence. Since SAM makes the loss landscape flatter, it helps interactive teaching to escape local optima. Finally, based on sharpness reduction, we propose a dual-level interactive strategy to further enhance performance and generalization, validating its effectiveness through experiments. In the future, we will further investigate the strategy design of interactive teaching in intelligent agents and consider how to reduce the complexity of the SAM algorithm in the context of interactive teaching.

## Acknowledgments and Disclosure of Funding

This work was supported by National Natural Science Foundation of China, Grant Number: 62476109, 62206108, and the Natural Science Foundation of Jilin Province, Grant Number: 20240101373JC, and Jilin Province Budgetary Capital Construction Fund Plan, Grant Number: 2024C008-5, and Research Project of Jilin Provincial Education Department, Grant Number: JJKH20241285KJ.

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

# A   Appendix A

Given independent observed data $\mathcal{D}$, hidden variable $Z$, and the probability model $p(\mathcal{D}, Z, \theta_f, \theta_g)$ with parameters $\theta_f$ and $\theta_g$, according to MLE, the optimal point estimate of $\theta_f$ is obtained when the likelihood of the model is maximized: $\theta_f = \arg\max_{\theta_f} p(\mathcal{D} \mid \theta_f, \theta_g)$. Suppose the training dataset is $\mathcal{D} = \{X_i = (x_i, y_i)\}_{i=1}^N$, where $x_i$ represents the input, $y_i$ represents the label, and noise is present. We define a hidden variable $Z_c \in \{0, 1\}$ to indicate whether the $c$-th sample is a noisy sample, where $Z_c = 0$ denotes a noisy sample and $Z_c = 1$ denotes a clean sample. $Z = Z_c^f \cup Z_c^g = \{Z_c\}_{c=1}^K$ represents the set of all sample hidden variables $Z_c$. The joint probability of $p(\mathcal{D}, Z_c | \theta_f, \theta_g)$ is obtained by simultaneously updating $f$ and $g$ for $X_i$ and $Z_c$. Considering the hidden variables, the likelihood of the discrete variable model can be expanded as follows:

$$p(\mathcal{D} \mid \theta_f, \theta_g) = \sum_{c=1}^K p(\mathcal{D}, Z_c \mid \theta_f, \theta_g), \quad Z = \{Z_1, \ldots, Z_K\}. \tag{18}$$

The hidden variable $Z$ represents any unobservable random variable in the probability model, and $p(\mathcal{D}, Z_c \mid \theta_f, \theta_g)$ is referred to as the joint likelihood of $\mathcal{D}$ and $Z_c$. Using the general approach of MLE and considering the independence of the observed data, we can take the logarithm of the equation above to obtain:

$$\log p(\mathcal{D} \mid \theta_f, \theta_g) = \log \prod_{i=1}^N p(X_i \mid \theta_f, \theta_g) = \sum_{i=1}^N \log p(X_i \mid \theta_f, \theta_g) = \sum_{i=1}^N \log \left[ \sum_{c=1}^K p(X_i, Z_c \mid \theta_f, \theta_g) \right].$$

Introducing a probability distribution $q(Z)$ related to the hidden variables, known as the latent distribution, which can be seen as the posterior of the hidden variables given the observed data. By applying Jensen's inequality $\log(\mathbb{E}[X]) \geq \mathbb{E}[\log(X)]$. The logarithm likelihood of the observed data $\mathcal{D}$ can be related as follows:

$$p(\mathcal{D} \mid \theta_f, \theta_g) = \sum_{i=1}^N \log \left[ \sum_{c=1}^K \frac{q(Z_c)}{q(Z_c)} p(X_i, Z_c \mid \theta_f, \theta_g) \right] \tag{19}$$

$$\geq \sum_{i=1}^N \sum_{c=1}^K \left[ q(Z_c) \log \frac{p(X_i, Z_c \mid \theta_f, \theta_g)}{q(Z_c)} \right] \equiv L(\theta_f, \theta_g, q), \tag{20}$$

where $L(\theta_f, \theta_g, q)$ is a lower bound for $\log p(\mathcal{D} \mid \theta_f, \theta_g)$. The EM algorithm [9] approximates the maximization of $\log p(\mathcal{D} \mid \theta_f, \theta_g)$ by maximizing this lower bound. When $\theta_f, \theta_g, q$ maximize the right-hand side of the inequality, the obtained $\theta_f, \theta_g$ at least yield a local maximum for the left-hand side of the inequality. Therefore, expressing the right-hand side as $\mathcal{L}(\theta_f, \theta_g, q)$, the EM algorithm aims to solve the following optimization problem:

$$\hat{\theta}_f, \hat{\theta}_g = \arg\max_{\theta_f, \theta_g} L(\theta_f, \theta_g, q), \tag{21}$$

where $L(\theta_f, \theta_g, q)$ is a lower bound for the MLE optimization problem, and the EM algorithm approximates [30] the maximum likelihood by maximizing a surrogate function. Specifically, the EM iteration process for the co-teaching algorithm can be expressed using the following proposition:

**Proposition A.1.** *Given the training dataset $\mathcal{D} = \{X_i = (x_i, y_i)\}_{i=1}^N$, which contains noisy samples, and assuming the samples are independent, we define the hidden variable $Z_c = 1$ to indicate that the $c$-th sample is a cleaner sample, meaning it has a lower loss value compared to noisy data. $Z = Z_c^f \cup Z_c^g = \{Z_c\}_{c=1}^K$ represents the set of hidden variables for all samples, and the corresponding latent distribution is denoted as $q(Z)$. The joint probability of $p(X_i, Z_c | \theta_f, \theta_g)$ is obtained by simultaneously updating neural networks $f$ and $g$ for $X_i$ and $Z_c$. The logarithm likelihood of the observed data $\mathcal{D}$ has the following lower bound $L$:*

$$L(\theta_f, \theta_g, q) \equiv \sum_{i=1}^N \sum_{c=1}^K \left[ q(Z_c) \log \frac{p(X_i, Z_c | \theta_f, \theta_g)}{q(Z_c)} \right], \tag{22}$$

*the EM algorithm approximates the maximization of $\log p(\mathcal{D} | \theta_f, \theta_g)$ by maximizing this lower bound $L(\theta_f, \theta_g, q)$. Specifically, the EM iteration process for the interactive teaching paradigm is as follows:*

$$Q(\theta_f, \theta_g^{(t)}) = \mathbb{E}_{Z_c^f | \mathcal{D}, \theta_f^{(t)}, \theta_g} [\log p(Z_c^f, \mathcal{D} | \theta_f, \theta_g^{(t)})], \tag{23}$$

$$Q(\theta_f^{(t)}, \theta_g) = \mathbb{E}_{Z_c^g | \mathcal{D}, \theta_f, \theta_g^{(t)}} [\log p(Z_c^g, \mathcal{D} | \theta_f^{(t)}, \theta_g)], \tag{24}$$

The subscript $Z_c^f | \mathcal{D}, \theta_f^{(t)}, \theta_g$ (or $Z_c^g | \mathcal{D}, \theta_f, \theta_g^{(t)}$) of the expectation represents the corresponding set of low-loss samples selected by network $f$ (or $g$) and is used to update network $g$ (or $f$).

**M-step:**

$$\theta_f^{(t+1)} = \arg\max_{\theta_f} Q(\theta_f^{(t)}, \theta_g), \tag{25}$$

$$\theta_g^{(t+1)} = \arg\max_{\theta_g} Q(\theta_f, \theta_g^{(t)}). \tag{26}$$

*Proof.* The EM algorithm in the interactive teaching process is a set of iterative computations, consisting of two steps: the E-step and the M-step. In the E-step, denoted as $E$, the previous iteration's values of $\theta_f^{(t)}$ (or $\theta_g^{(t)}$) are fixed, and the posterior latent distribution of $q_f^{(t+1)}$ (or $q_g^{(t+1)}$) with respect to $L(\theta_f, \theta_g, q)$ is calculated. In the M-step, denoted as $M$, the weights of the network parameters $\theta_f^{(t+1)}$ (or $\theta_g^{(t+1)}$) are updated using $q_g^{(t+1)}$ (or $q_f^{(t+1)}$) to maximize lower bound $L(\theta_f, \theta_g, q)$. The interactive teaching starts iterating after initializing the model parameters, with the E-step and M-step alternating during each iteration. This aligns with the consistent update process of general EM algorithm. The following outlines the derivation of the E-step and M-step of the EM algorithm about the interactive teaching update process.

**1. E-step (Expectation-step)**

According to the objective of the EM algorithm, the E-step involves computing the latent variable distribution $q$ that maximizes the lower bound, given the fixed network parameters $\theta_f$ (or $\theta_g$) computed in the previous step, the lower bound is as follows $L$:

$$q = \arg\max_q L(\theta_f, \theta_g, q) = \arg\max_q \sum_{i=1}^{N} \sum_{c=1}^{K} \left[ q(Z_c) \log \frac{p(X_i, Z_c | \theta_f, \theta_g)}{q(Z_c)} \right]. \tag{27}$$

Taking into account the previous Inequality 20, $\log p(\mathcal{D} | \theta_f, \theta_g) - L(\theta_f, \theta_g, q)$ as follows:

$$\log p(\mathcal{D} | \theta_f, \theta_g) - L(\theta_f, \theta_g, q)$$

$$= \sum_{i=1}^{N} \log \left[ \sum_{c=1}^{K} p(X_i, Z_c | \theta_f, \theta_g) \right] - \sum_{i=1}^{N} \sum_{c=1}^{K} \left[ q(Z_c) \log \frac{p(X_i, Z_c | \theta_f, \theta_g)}{q(Z_c)} \right]$$

$$= \sum_{i=1}^{N} \left[ \log p(X_i | \theta_f, \theta_g) \sum_{c=1}^{K} q(Z_c) - \sum_{c=1}^{K} q(Z_c) \log \frac{p(X_i, Z_c | \theta_f, \theta_g)}{q(Z_c)} \right]$$

$$= \sum_{i=1}^{N} \sum_{c=1}^{K} q(Z_c) \left[ \log p(X_i | \theta_f, \theta_g) - \log \frac{p(X_i, Z_c | \theta_f, \theta_g)}{q(Z_c)} \right]$$

$$= \sum_{i=1}^{N} \sum_{c=1}^{K} q(Z_c) \log \left[ \frac{p(X_i | \theta_f, \theta_g) q(Z_c)}{p(X_i, Z_c | \theta_f, \theta_g)} \right],$$

$$\sum_{i=1}^{N} \sum_{c=1}^{K} q(Z_c) \log \left[ \frac{q(Z_c)}{p(Z_c | X_i, \theta_f, \theta_g)} \right] = \sum_{i=1}^{N} \mathrm{KL}\left[ q(Z) \| p(Z | X_i, \theta_f, \theta_g) \right], \tag{28}$$

$$\Rightarrow L(\theta_f, \theta_g, q) = \log p(\mathcal{D} | \theta_f, \theta_g) - \sum_{i=1}^{N} \mathrm{KL}\left[ q(Z) \| p(Z | X_i, \theta_f, \theta_g) \right], \tag{29}$$

where KL represents the Kullback-Leibler divergence. Based on the properties of KL divergence, its minimum value is achieved when the two probability distributions are equal. Therefore, when

$q(Z) = p(Z \mid \mathcal{D}, \theta_f, \theta_g)$, the lower bound $L(\theta_f, \theta_g, q)$ is maximized. For the $t$-th iteration, the E-step is computed as follows, taking into account that $\sum_{c=1}^{K} q(Z_c) = 1$, by applying Bayes' theorem, the above equation can be transformed into:

$$\max_q L(\theta_f, \theta_g, q) \iff \min_q \sum_{i=1}^{N} \mathrm{KL}\left[q(Z) \| p(Z \mid X_i, \theta_f, \theta_g)\right], \tag{30}$$

$$q_f^{(t+1)} = p\left(Z_c^f \mid \mathcal{D}, \theta_f^{(t)}, \theta_g\right), \; q_g^{(t+1)} = p\left(Z_c^g \mid \mathcal{D}, \theta_f, \theta_g^{(t)}\right). \tag{31}$$

At this point, we have completed the calculation of the posterior distribution of latent variables in the E-step of the EM algorithm, as described in Equation 31.

### 2. M-step (Maximization step)

Building upon the E-step, the M-step involves solving for model parameters that maximize $L(\theta_f, \theta_g, q)$. For network $f$, the necessary condition for this extremum problem is $\partial L(\theta_f, \theta_g, q) / \partial \theta_f = 0$:

$$\max_{\theta_f} L(\theta_f, \theta_g, q) \Rightarrow \frac{\partial}{\partial \theta_f}[L(\theta_f, \theta_g, q)] = 0, \tag{32}$$

$$\Rightarrow \frac{\partial}{\partial \theta_f}\left[\sum_{i=1}^{N}\sum_{c=1}^{K} q(Z_c) \log p(X_i, Z_c \mid \theta_f, \theta_g)\right] = 0, \tag{33}$$

$$\Rightarrow \frac{\partial}{\partial \theta_f} \mathbb{E}_q[\log p(\mathcal{D}, Z \mid \theta_f, \theta_g)] = 0, \tag{34}$$

where $\mathbb{E}_q$ represents the mathematic expectation of the joint likelihood $p(\mathcal{D}, Z \mid \theta_f, \theta_g)$ with respect to the hidden distribution $q(Z)$. Based on this, the computation for the M-step in interactive teaching is obtained as follows:

$$\theta_f^{(t+1)} = \arg\max_{\theta_f} \mathbb{E}_{q_g^{(t)}}[\log p(Z_c^g, \mathcal{D} \mid \theta_f^{(t)}, \theta_g)], \tag{35}$$

$$\theta_g^{(t+1)} = \arg\max_{\theta_f} \mathbb{E}_{q_f^{(t)}}[\log p(Z_c^f, \mathcal{D} \mid \theta_f, \theta_g^{(t)})]. \tag{36}$$

For network $g$, the derivation follows the same process as described above. Maximum likelihood estimation is equivalent to minimizing empirical risk in machine learning. $\qquad\square$

## B   Appendix B

### B.1   Figures on datasets

We present the test accuracy figures for different noise types across MNIST, FMNIST,CIFAR10, SVHN, CIFAR100, with a noise level of 20%. It is evident from the figures that Sharpness Reduction Interactive Teaching (SRIT) exhibits superior generalization performance and is less prone to overfitting.

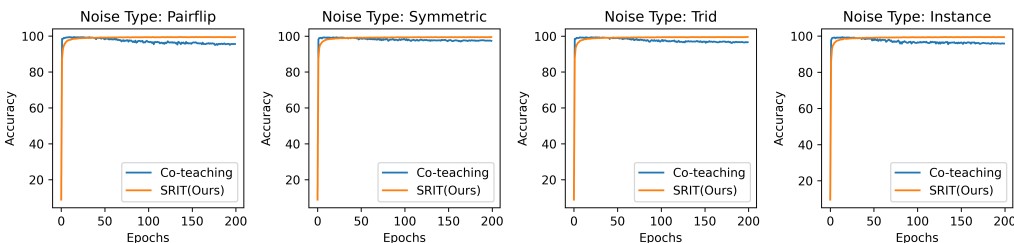

Figure 4: Plotting charts based on the type of noise, with a noise rate of 20%. The charts compare the test performance of co-teaching and Sharpness Reduction Interactive Teaching (SRIT) on the MNIST dataset.

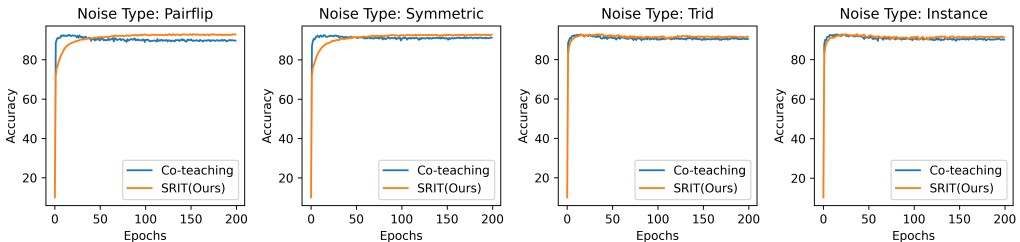

Figure 5: Figures based on the type of noise, with a noise rate of 20%. The charts compare the test performance of co-teaching and Sharpness Reduction Interactive Teaching (SRIT) on the FMNIST dataset.

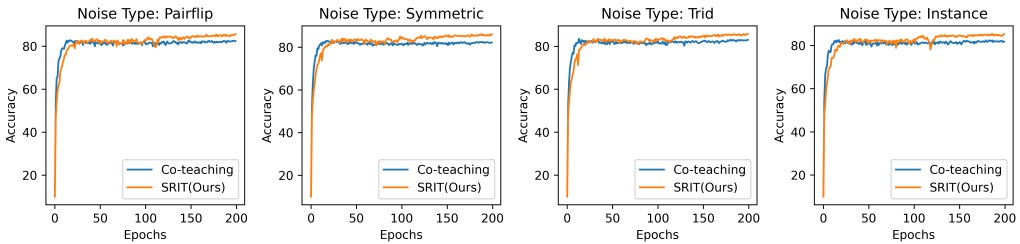

Figure 6: Figures based on the type of noise, with a noise rate of 20%. The charts compare the test performance of co-teaching and Sharpness Reduction Interactive Teaching (SRIT) on the CIFAR10 dataset.

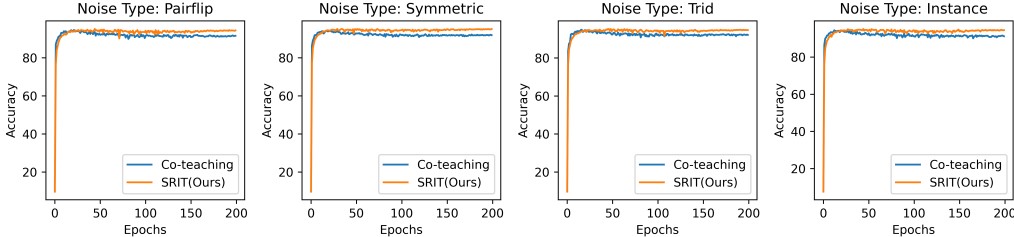

Figure 7: Figures based on the type of noise, with a noise rate of 20%. The charts compare the test performance of co-teaching and Sharpness Reduction Interactive Teaching (SRIT) on the SVHN dataset.

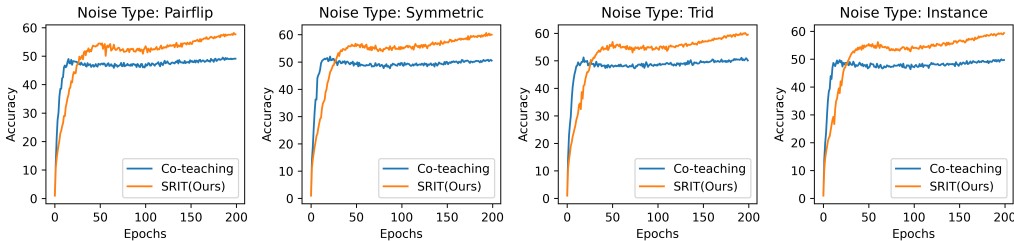

Figure 8: Figures based on the type of noise, with a noise rate of 20%. The charts compare the test performance of co-teaching and Sharpness Reduction Interactive Teaching (SRIT) on the CIFAR100 dataset.

