# OpenReview forum: "Sharpness-Aware Minimization Activates the Interactive Teaching's Understanding and Optimization"
_NeurIPS.cc/2024/Conference — NeurIPS 2024 poster_

### Official Review · Reviewer_d2nj · 2024-07-08

**Soundness:** 3
**Presentation:** 3
**Contribution:** 3
**Rating:** 5
**Confidence:** 4

**Summary:**

The authors investigate the parameter update mechanism of interactive teaching, represented by co-teaching, in their article. They provide an algorithmic understanding from the perspective of Expectation-Maximization (EM) and utilize SAM (Sharpness-Aware Minimization) for interactive teaching optimization from the loss landscape perspective. The authors demonstrate the effectiveness of their approach through experiments.

**Strengths:**

The authors present a cohesive logical framework with a clear structure in their article. The accomplished work provides insightful implications for further exploration in the domain of interactive teaching tasks.

**Weaknesses:**

1. The description of the experimental section in the article is somewhat weak, and the authors need to present the relationship between the baselines in the experiment. As mentioned in the paper, there are many methods with the theme of interactive teaching, such as co-teaching+, JoCoR, CNLCU. Why were experiments only conducted on co-teaching and CNLCU?
2. The authors' choice of using co-teaching as the backbone for studying interactive teaching raises questions about its representativeness and persuasiveness as a technique.
3. The author's description of the E-step update in Proposition 4.1 is not clear enough.

**Questions:**

see weakness.

**Limitations:**

The limitations are briefly discussed in the paper.

---

> ### Author Rebuttal · Authors · 2024-08-04
>
> Thank you for your comments and for your interest in the experimental section. We will explain the rationale behind the experiments and provide details on updating certain formulas.
>
> **Q1**: The description of the experimental section in the article is somewhat weak, and the authors need to present the relationship between the baselines in the experiment. As mentioned in the paper, there are many methods with the theme of interactive teaching, such as co-teaching+, JoCoR, CNLCU. Why were experiments only conducted on co-teaching and CNLCU?
>
> **A**: Co-teaching, co-teaching+[1], JoCoR, and CNLCU were developed sequentially over time. Therefore, during the experimental process, we chose to conduct experiments on the most classic version of co-teaching and the recent CNLCU. Additionally, methods used in MentorNet[2] differ significantly from the co-teaching series. MentorNet trains an additional network to filter out noisy data, thereby providing sample weighting for the StudentNet. This approach differs from the analysis of the co-teaching algorithm in this paper from the perspectives of the loss landscape and EM. Hence, we opted to conduct multiple experiments on representative versions of co-teaching and CNLCU.
>
> **Q2**: The authors' choice of using co-teaching as the backbone for studying interactive teaching raises questions about its representativeness and persuasiveness as a technique.
>
> **A**: Strictly speaking, co-teaching cannot fully represent interactive teaching. Due to the diversity of research, various teaching strategies have their own advantages and characteristics, making it difficult to find an algorithm that can comprehensively cover the vast topic of interactive teaching. We acknowledge that co-teaching is just one of many interactive teaching methods, albeit the most representative one. Apart from co-teaching and its direct extensions, there have been numerous research efforts inspired by the co-teaching framework, such as JoCoR, co-learning[3], and CNLCU. Therefore, a deeper understanding of interactive teaching algorithms and research on optimizing generalization still require further exploration based on the specific settings of each algorithm.
>
> **Q3**: The author's description of the E-step update in Proposition 4.1 is not clear enough.
>
> **A**: In the E-step of Proposition 4.1, taking Equation (13) as an example, the left side of the equation $Q(\theta_f, \theta_g^{(t)})$ represents the $g$ network with fixed parameters of network $f$, where $\theta_f$ can be considered a constant. These parameters are then inputted into Equation (16) for updating the parameters of network $g$. On the right side of Equation (13), it represents an expected log-likelihood. The latent variable data distribution $Z^{f}_c$ pertains to the data distribution selected from network $f$. The logarithmic term indicates the probability of network $g$ occurring under the data distribution $Z^{f}_c$.
>
> Reference:
>
> [1] Yu X, Han B, Yao J, Niu G, Tsang I, Sugiyama M. How does disagreement help generalization against label corruption?. InInternational conference on machine learning 2019 May 24 (pp. 7164-7173). PMLR.
>
> [2] Jiang L, Zhou Z, Leung T, Li LJ, Fei-Fei L. Mentornet: Learning data-driven curriculum for very deep neural networks on corrupted labels. InInternational conference on machine learning 2018 Jul 3 (pp. 2304-2313). PMLR.
>
> [3] Tan C, Xia J, Wu L, Li SZ. Co-learning: Learning from noisy labels with self-supervision. InProceedings of the 29th ACM International Conference on Multimedia 2021 Oct 17 (pp. 1405-1413).

---

> > ### Comment · Reviewer_d2nj · 2024-08-10
> > **Response**
> >
> > I read the replies, and most of my concerns have been resolved. Therefore, I have raised my score.

---

### Official Review · Reviewer_4Qn5 · 2024-07-08

**Soundness:** 3
**Presentation:** 3
**Contribution:** 3
**Rating:** 5
**Confidence:** 4

**Summary:**

This paper delves into the understanding and optimization of teaching strategies between artificial intelligence agents. Taking co-teaching as an example, the authors propose an understanding of co-teaching based on the EM algorithm within a probabilistic framework. Building on this, they point out that co-teaching can get stuck in local optima during the optimization update process. To address this, they propose the use of Sharpness Aware Minimization (SAM) to make the loss surface smoother and improve generalization performance.

**Strengths:**

The paper provides a clear description of the explored problem. The authors offer a probabilistic understanding of heuristic interactive teaching strategies. Leveraging SAM, they propose a solution and optimization method, achieving performance improvements in their experiments.

**Weaknesses:**

1 The paper explores the understanding of interactive teaching from a probabilistic perspective, using co-teaching as an example. To introduce the EM framework, the authors assume the cleanliness of the data distribution is unknown and treat it as a latent variable. However, as co-teaching progresses and reduces the noise in the data, the distribution of this latent variable is likely to change. It remains unclear whether the EM process would still hold under such circumstances.
2 The description of Proposition 4.1 in the paper remains questionable and lacks clarity. A concise explanation of the formula in the M step needs to be provided in the proposition.
3 The authors use Proposition 4.1 with a probabilistic model to explain coteaching, making strong assumptions about certain concepts, such as the jointprobability $p(X_i, Z_c | \theta_f, \theta_g)$.
4 The complexity of Algorithm 1 still needs to be optimized.

**Questions:**

see weakness.

**Limitations:**

The limiations are discussed in the paper.

---

> ### Author Rebuttal · Authors · 2024-08-04
>
> Thank you for your thorough comments on this paper. Based on the comments provided, we will offer specific explanations.
>
> **Q1**: The paper explores the understanding of interactive teaching from a probabilistic perspective, using co-teaching as an example. To introduce the EM framework, the authors assume the cleanliness of the data distribution is unknown and treat it as a latent variable. However, as co-teaching progresses and reduces the noise in the data, the distribution of this latent variable is likely to change. It remains unclear whether the EM process would still hold under such circumstances.
>
> **A**: In traditional EM algorithm processes, the distribution of latent variables remains unchanged. However, in co-teaching, as the training progresses, the selected data points tend to have fewer clean data points to reduce the adverse effects of noisy data on the model in the later stages of training. An intuitive analysis reveals that the data distribution of latent variables changes. Initially, a larger number of data points with small loss values are selected, and when the model performance is not sufficiently robust, the data distribution of latent variables contains more noisy data. Subsequently, more clean samples are included, reducing the diversity of samples and potentially leading to overfitting.
>
> This change in data distribution does not affect the subsequent update process. In extreme cases where only clean samples are selected, the EM process updated by co-teaching will operate on the same latent variable distribution rather than on different distributions as seen earlier.
>
> **Q2**: The description of Proposition 4.1 in the paper remains questionable and lacks clarity. A concise explanation of the formula in the M step needs to be provided in the proposition.
>
> **A**: In the E-step, for example, the right side of Equation (13) indicates the use of the $f$ network to select training data for the training process of the $g$ network, where $\theta_g^{(t)}$ represents the involvement of the $g$ network in computations. On the left side of Equation (13), $Q(\theta_f, \theta_g^{(t)})$ corresponds to Equation (16), signifying the parameter update of the $g$ network under the existing data selected by the $f$ network. Reviewers can also refer to Remark 4.2 and the appendix section, where the parameter update process is conducted iteratively.
>
> **Q3**: The authors use Proposition 4.1 with a probabilistic model to explain coteaching, making strong assumptions about certain concepts, such as the joint probability $p(X_i,Z_c|\theta_f,\theta_g)$.
>
> **A**: We have indeed made a strong assumption, which is typically assumed in probability theory that there is always a certain gap between the data distribution and the actual data distribution. However, we believe there is merit to this assumption. In the joint probability $p(X_i,Z_c|\theta_f,\theta_g)$, we consider that in the presence of both networks $f$ and $g$, the training data distribution $X_i$ and the latent variable data distribution $Z_c$ occur simultaneously. There is a certain overlap in the data distributions between $X_i$ and $Z_c$.
>
> **Q4**: The complexity of Algorithm 1 still needs to be optimized.
>
> **A**:  As pointed out by other reviewers, the complexity of Algorithm 1 is indeed high due to involving computations of two networks and two gradient calculations in the SAM method.  However, we want to emphasize that the key aspects of our approach are:
>
> 1) Experimental results show that compared to the original method, using the SAM method significantly enhances generalization performance on noisy data, for example, on MNIST with a Noise Ratio of 0.4, the accuracy increased from [91%, 94%] to [98%, 99%], which is a reasonable cost for accuracy improvement calculations.
>
> 2) It has a linear increase in time complexity, increasing linearly only with the input size.
>
> In special environments with limited computational resources such as embedded systems, and edge computing, sacrificing some computational accuracy may be necessary. In such cases, methods like LookSAM, which reduce complexity, should be considered. We will consider optimizing the computational complexity of SAM method in interactive teaching scenarios like co-teaching in the future.

---

> > ### Comment · Reviewer_4Qn5 · 2024-08-10
> >
> > Thanks for the clear clarification. The response addresses the concerns about the EM process and the Proposition 4.1. I have no other questions and I'd love to raise my score

---

### Official Review · Reviewer_hGPy · 2024-07-12

**Soundness:** 3
**Presentation:** 4
**Contribution:** 3
**Rating:** 5
**Confidence:** 4

**Summary:**

In this paper, the author focus on the research of teaching method, choosing the well-known Co-teaching as a prototype to make deeper analysis. Co-teaching simultaneously optimizes dual networks and both networks select small-loss samples for the other larger loss landscape reduction. On the other hand, Sharpness-Aware Minimization (SAM) learns the model on a flatter loss landscape, which also provides with robustness and stability, and can be well-combined with Co-teaching. Besides, the author further explain the learning process of Co-teaching in an EM aspect. Sufficient experiments are completed to demonstrate the superior performance of the proposed method.

**Strengths:**

The strengths of the paper are listed as follows,

1. The paper is quite well-written and easy to follow, the motivation from teaching aspect introduced in Intro is interesting.
2. The chosen teaching prototype Co-teaching select small-loss samples for reducing larger loss landscape, while SAM provides flatter loss landscape. Such combination is moderate and well-explained in Fig.2.
3. The experiments are done sufficiently on five datasets and different data noises to prove the performance.

**Weaknesses:**

The weakness of this paper are listed as following,

1. From my perspective, Co-teaching and SAM can be well combined from loss landscape aspect, which has been explained in Section 4.1. On the other side, the EM algorithm seems to be an elucidation of Co-teaching method, which seems dissected with the former sections. I wonder if the author can give more explanation, or the section of EM might be optimized and replaced by sections making further optimization to Co-teaching.
2. About the proposed method, what I most concern is the computational cost. Co-teaching cooperates dual networks to select cleaner data samples for training, which has demanded for doubled computing time. Besides, SAM calculates losses before and after perturbation, which also requires additional compute cost. I wonder the author can provides with time cost ablation on training data, comparing to Co-teaching itself and other NLL methods.

**Questions:**

Please refer to the weaknesses.

**Limitations:**

The proposed method is aimed to combat training data with noise. However, the author didn't offer a section claiming limitations.

---

> ### Author Rebuttal · Authors · 2024-08-04
>
> Thank you for your constructive question. In our response below, we further explain the motivation behind the article and provide additional experiments regarding the time complexity.
>
> **Q1**:  From my perspective, Co-teaching and SAM can be well combined from loss landscape aspect, which has been explained in Section 4.1. On the other side, the EM algorithm seems to be an elucidation of Co-teaching method, which seems dissected with the former sections. I wonder if the author can give more explanation, or the section of EM might be optimized and replaced by sections making further optimization to Co-teaching.
>
> **A**:  Regarding the description in the fourth section, there may be some confusing points that require further clarification and elaboration.
>
> - The title of the paper is "Sharpness-Aware Minimization Activated Interactive Teaching Understanding and Optimization". Taking co-teaching as an example, inspired by SAM, we consider the actual optimization process from the perspective of the loss landscape. The preliminary conclusion we arrive at is that co-teaching eliminates data points with large loss values, which is reflected in the loss landscape as a reduction in the surfaces with high loss values.
>
> - However, in practical experiments, co-teaching tends to overfit in the later stages of training. At this point, we find that the co-teaching process can be modeled probabilistically using a variant of the EM algorithm. The goal of co-teaching is to select clean data, while EM models clean data as latent variables. Additionally, EM easily falls into local optima, which is a primary cause of overfitting. On the other hand, SAM method demonstrates good generalization by optimizing sharp points on surfaces, reducing surface curvature, and enhancing performance.
>
> - Therefore, introducing SAM into the optimization of co-teaching helps in obtaining flatter loss surfaces, making it easier to escape local optima. These correspond to the sections in the paper as follows: Section 4.1 introduces the optimization process of co-teaching's loss landscape, Section 4.2 describes the EM parameter update process of co-teaching, and Section 4.3 details the specific algorithmic process of integrating SAM into co-teaching.
>
> **Q2**:  About the proposed method, what I most concern is the computational cost. Co-teaching cooperates dual networks to select cleaner data samples for training, which has demanded for doubled computing time. Besides, SAM calculates losses before and after perturbation, which also requires additional compute cost. I wonder the author can provides with time cost ablation on training data, comparing to Co-teaching itself and other NLL methods.
>
> **A**:  Due to the inherent property of requiring two gradient computations, using SAM results in higher complexity compared to general interactive teaching methods such as co-teaching. Experimental results from co-teaching and CNLCU show that employing the SAM method significantly enhances generalization performance on noisy data. While introducing SAM does indeed increase computational time, this is a reasonable cost paid for achieving better generalization performance. For example, on MNIST with a Noise Ratio of 0.4, the accuracy increased from [91%, 94%] to [98%, 99%], on CIFAR100 with a Noise Ratio of 0.2, the accuracy increased from [48%, 50%] to [57%, 59%]. We believe that this increase in computational cost is justified as it notably improves the stability and reliability of the model in noisy environments. The increase in computational time complexity is linear, and when dealing with larger amounts of data, it does not exponentially increase computational load.
>
> In response to the experimental proposals from the reviewers, we have conducted disintegration experiments regarding the time cost of training data and will provide additional information in the subsequent appendix.
>
> Time expenditure (S) on $\text{MNIST,CIFAR10,CIFAR100}$
> | Noise type | Symmetric. | Pairflip. | Tridiagonal. | Instance. |
> | :---: | :---: | :---: | :---: | :---: |
> | MNIST, Noise ratio $=0.2$  |
> | Co-teaching | 5991.20 | 5845.71 | 5758.94 | 5835.25 |
> | Co-teaching + SAM | 12447.52 | 12364.43 | 125569.80 | 12751.16|
> | CIFAR10, Noise ratio $=0.4$ |
> | Co-teaching | 6292.33 | 6416.06 | 6356.10 | 6436.37 |
> | Co-teaching + SAM | 131391.94 | 13860.54 | 13873.92 |13945.70|
> | CIFAR 100 , Noise ratio $=0.2$|
> | Co-teaching | 6524.65 | 6356.50 | 6381.92 | 6321.79 |
> | Co-teaching + SAM | 13903.19 | 13949.09  | 13727.83 | 13825.99 |
>
> In the presence of NVIDIA RTX 6000 GPUs with 24GB of memory, we record the computational times of certain experiments. Each GPU runs a subprocess independently. In the computations for co-teaching, we accounted for the additional computational time introduced by incorporating the SAM method. By randomly selecting two different noise ratios under the MNIST, CIFAR10, and CIFAR100 datasets, along with four noise types, and calculating algorithm running times, upon analyzing the table "Time expenditure (S) on MNIST,CIFAR10,CIFAR100". We observe that for each dataset and noise ratio combination, the computational time with SAM added typically doubles compared to the co-teaching method without SAM. This aligns with our analysis that SAM requires an extra gradient calculation. Furthermore, the computational time on MNIST is slightly lower than that on CIFAR10 and CIFAR100. The computation times for CIFAR10 and CIFAR100 are comparable.

---

> > ### Comment · Reviewer_hGPy · 2024-08-10
> >
> > Thank you for your detailed response, but I'm sorry to inform that I must adjust my rating. My primary concern remains with the computational cost. The reported time expenditure is higher than what I expected, accordingly, **Co-teaching + SAM requires more than doubled time than Co-teaching alone. Notably, Co-teaching itself is already a dual network.** Though your method demonstrates clear improvement on robustness, the over four times increase in computational cost than the vanilla network is somekind unacceptable from my perspective. Anyway, I appreciate the clarity and quality of writing, and maintain a positive rating score.

---

> > > ### Author Response · Authors · 2024-08-10
> > >
> > > Thank you for your reply. There are some unclear points and certain misunderstandings in it, which I would like to clarify:
> > >
> > >  1. The topic we are exploring is interactive teaching, which typically involves two or more models, making the complexity of our algorithm naturally higher than that of a single model. The essence of the Sharpness-Aware Minimization (SAM) method is a more refined exploration of the gradient space, implicitly utilizing second-order gradient information, significantly improving generalization performance in experiments. Some additional computations are inevitable but not significant. Furthermore, compared to directly computing the eigenvalues or trace of the Hessian matrix to enhance generalization performance, SAM has greatly reduced computational costs and complexity.
> > >
> > >  2. In terms of generalization performance, when expanding to a larger number of models or models with more parameters, using SAM leads to better gradient processing and parameter aggregation effects. This outperforms solely using co-teaching, and the additional gradient computation costs do not pose a significant obstacle to the improvement in generalization performance.
> > >
> > >  3. From a hardware resource perspective, the computational power consumption of the current algorithm complexity can be kept within a reasonable range without significant scalability changes. The increase in complexity compared to computational resources is tolerable.
> > >
> > >  4. From a technical standpoint, considering utilizing tricks like low-rank matrix decomposition, low-rank adaptation, and other existing technical tools to reduce complexity is feasible. This is not difficult, and complexity arising from gradient computations will no longer be the core issue，it is not the main point of concern.
> > >
> > > Once again, thank you for reviewing our paper.

---

> > > ### Author Response · Authors · 2024-08-13
> > >
> > > We hope that our response can objectively reflect our perspective on this paper. We need to emphasize that compared to other second-order optimization methods such as directly computing the eigenvalues or traces of the Hessian matrix to enhance generalization performance, methods like SAM have already reduced computational costs and complexity.  If the complexity of the second-order method based on the Hessian is $O(n^2)$, our complexity is $O(2n)$. Nevertheless, discussions about complexity are not within the core scope of this paper.
> > >
> > > - The byproduct of the optimization: complexity issues but not primary
> > >
> > >   1). **Complexity Analysis**: The essence of the SAM method lies in a more refined exploration of the gradient space, implicitly utilizing second-order information about the parameter space in the loss landscape. In experiments, it significantly improves generalization performance but incurs some unavoidable additional computations. The computational power consumption of the current algorithm complexity can be kept within a reasonable range and does not exponentially increase with the scale of data and models. This increase in complexity, compared to computational resources, is considered acceptable.
> > >
> > >   2). **Expansion and Solutions**: When scaling to more models or models with larger parameter sizes, SAM's gradient handling and parameter aggregation effects will be more beneficial than solely using co-teaching. The additional cost of gradient computations only linearly increases, which does not hinder the improvement in generalization performance. From a technical standpoint, considering the use of low-rank tricks and other existing technical tools can help reduce complexity, shifting complexity away from gradient computations as the core issue.

---

### Official Review · Reviewer_N9mR · 2024-07-15

**Soundness:** 3
**Presentation:** 3
**Contribution:** 3
**Rating:** 7
**Confidence:** 4

**Summary:**

This work mainly proposes an understanding of the interactive teaching algorithm update mechanism led by co-teaching, as well as further optimization using the sharpness technique. The article proceeds from two aspects: one is to summarize that the co-teaching update process can be explained by the EM method, and the second is to enhance generalization with the help of the SAM algorithm.

**Strengths:**

The paper is clear and easy to read, with a complete and logical argumentation. The authors attempt to understand the co-teaching training process using the principles of the EM method. Due to the limitations of the EM method, they use the SAM method to improve the experimental results.

**Weaknesses:**

1. The authors' arguments and experiments are only on the interactive teaching of two neural networks. If conducted on three or even more networks, it is unclear whether the insights provided in the paper would still hold. The generalization capability on three or more intelligent agents remains unknown.
2.In terms of generalization, the authors argue that SAM can flatten the loss landscape of co-teaching, making it less prone to local optima and thus improving generalization. They provide empirical evidence in the paper, but it should be noted that this approach increases the complexity of the algorithm. SAM requires computing parameters twice for a single network update, while co-teaching already involves interaction between two networks.

**Questions:**

See above

**Limitations:**

See above

---

> ### Author Rebuttal · Authors · 2024-08-04
>
> Thank you for your thoughtful question. We provide the following explanation in response to your question.
>
> **Q1**: The authors' arguments and experiments are only on the interactive teaching of two neural networks. If conducted on three or even more networks, it is unclear whether the insights provided in the paper would still hold. The generalization capability on three or more intelligent agents remains unknown.
>
> **A**: In the current trend of language model-based agents learning, we have witnessed the extensive research value of collaborative interactive learning. In the existing research on Interactive teaching, such as Co-training[1], MentorNet[2], and Decoupling[3], typically only involve knowledge exchange between two networks. Multiple networks interactions are often built upon the foundation of interactions between two networks. When training interactions involve three or more neural networks, it may entail more complex tasks, requiring strategies designed according to practical contexts. We believe that in interactions involving three or more networks, complex tasks often need to be decomposed into pairwise interactions, adopting a divide-and-conquer strategy, and then judiciously merging them. In summary, we reasonably infer that the generalization of multiple networks interactions still maintains the properties observed with two networks.
>
> **Q2**:  In terms of generalization, the authors argue that SAM can flatten the loss landscape of co-teaching, making it less prone to local optima and thus improving generalization. They provide empirical evidence in the paper, but it should be noted that this approach increases the complexity of the algorithm. SAM requires computing parameters twice for a single network update, while co-teaching already involves interaction between two networks.
>
> **A**:  Due to the inherent property of requiring two gradient computations, using SAM involves higher complexity compared to conventional interactive teaching methods. Experimental results demonstrate a significant enhancement in generalization performance on noisy data with the use of SAM. This is a reasonable cost paid to improve accuracy computation, as it only incurs a linear increase in time complexity.
>
> When dealing with multiple networks computations, it does not exponentially increase the computational load, but rather linearly adds complexity. However, it is important to objectively point out that in interactive teaching, multiple networks and multiple intelligent agents often need to ultimately resort to pairwise computations. In special environments with limited computational resources, such as edge computing, computational complexity cannot be ignored. In such cases, sacrificing a portion of computational accuracy and considering complexity-reducing methods like LookSAM [4] becomes necessary, but this is beyond the scope of this paper's discussion.
>
> [1] Blum A, Mitchell T. Combining labeled and unlabeled data with co-training. InProceedings of the eleventh annual conference on Computational learning theory 1998 Jul 24 (pp. 92-100).
>
> [2] Jiang L, Zhou Z, Leung T, Li LJ, Fei-Fei L. Mentornet: Learning data-driven curriculum for very deep neural networks on corrupted labels. InInternational conference on machine learning 2018 Jul 3 (pp. 2304-2313). PMLR.
>
> [3] Malach E, Shalev-Shwartz S. Decoupling" when to update" from" how to update". Advances in neural information processing systems. 2017;30.
>
> [4] Liu Y, Mai S, Chen X, Hsieh CJ, You Y. Towards efficient and scalable sharpness-aware minimization. InProceedings of the IEEE/CVF Conference on Computer Vision and Pattern Recognition 2022 (pp. 12360-12370).

---

> > ### Comment · Reviewer_N9mR · 2024-08-13
> > **To Authors**
> >
> > Thank you for your detailed rebuttal, which solved most of my concerns. I am willing to raise the rating to weak accept.

---

### Author Rebuttal · Authors · 2024-08-04

We thank all the reviewers for their detailed questions regarding this paper. We acknowledge that this work still requires additional clarifications and explanations, broadly in three areas: **core logic**, **experiments and complexity**, and **explanations of certain principles**. We provide the following general comments and feedback for all reviewers and the area chair.

- Firstly, regarding the core logic of the paper, it can be understood from three perspectives.

  (1). **Motivation.**  From the loss landscape perspective, we believe that the co-teaching strategy, which involves interactive teaching through the selection of small-loss data points, reduces the high-loss plane. This is inspired by the SAM (Sharpness-Aware Minimization) method, which optimizes the sharp points of the loss plane to make the loss landscape flatter.

  (2). **Insights.**  We offer an understanding of the co-teaching update steps as an EM (Expectation-Maximization) parameter update process. This is based on a probabilistic view, assuming noisy data as latent variables and iteratively updating parameters through mutual interaction. However, the EM algorithm has inherent drawbacks, such as easily getting stuck in local optima.

  (3). **Solutions.**  Since co-teaching does not alter the curvature of the loss landscape, incorporating the SAM method makes the loss plane flatter, enabling parameter updates to more easily escape local minima. Therefore, SAM can further optimize the co-teaching and other teaching methods.

- Secondly, in terms of experiments, we additionally provide representative experimental results to indicate that there is an increase in computation time after using SAM. This is an intrinsic property of the SAM method, which involves computing gradients twice.

   (1). **Experiment.**  Our experimental results show that the use of SAM significantly improves the performance of the co-teaching method on noisy data. For example, on MNIST with a Noise Ratio of 0.4, the accuracy increased from [91%, 94%] to [98%, 99%], on CIFAR100 with a Noise Ratio of 0.2, the accuracy increased from [48%, 50%] to [57%, 59%].

   (2). **Analysis.**  These results indicate that the SAM method has significant advantages in enhancing model generalization. While the introduction of the SAM method does increase computational time, this is a reasonable cost paid for achieving better generalization performance. It has a linear increase in time complexity, and the additional gradient computations do not exponentially rise with the increase in input size; they only linearly increase in complexity in terms of gradient calculations. We believe that this increase in computational cost is worthwhile as it significantly improves the stability and generalization of the model in noisy environments.

 - Finally, in the explanation provided for the formula updating process in Proposition 4.1, we elucidate the differences of the co-teaching cross-iterative parameter process from conventional gradient updating methods. Therefore, when describing it using the EM framework, we highlight the details of interactive iterative parameters in Proposition 4.1, which distinguishes it from the classical EM method.

---

> ### Author Response · Authors · 2024-08-13
>
> We hope that our response can objectively reflect our perspective on this paper. We need to emphasize that compared to other second-order optimization methods such as directly computing the eigenvalues or traces of the Hessian matrix to enhance generalization performance, methods like SAM have already reduced computational costs and complexity.  If the complexity of the second-order method based on the Hessian is $O(n^2)$, our complexity is $O(2n)$. Nevertheless, discussions about complexity are not within the core scope of this paper.
>
> - The core idea of this paper can be summarized as follows:
>
>   1). **Characteristics of Loss Landscape**: Two network interactive teaching methods represented by co-teaching, retain the small loss data values from each iteration and pass them to the other network for parameter updates. From the perspective of loss landscape, this entails updating parameters of the network on the small loss surface of the other. In the paper, we make reasonable hidden variable assumptions and use the EM algorithm to summarize this process interactively. Through analysis, we believe that while the co-teaching method enhances robustness, it is prone to getting stuck in local optima, which aligns with the local optimality property of EM.
>
>   2). **Optimization of Loss Landscape**: The performance enhancement method represented by SAM, by optimizing the large curvature loss landscape, can effectively alleviate the issue of the aforementioned co-teaching method being prone to local optima and lacking in generalization capability. In co-teaching, it interacts to optimize the other's loss landscape, introduces perturbations in parameter space, selects the direction of fastest gradient descent for parameter updates, flattening the loss landscape and enhancing the performance and generalization of co-teaching.
>
> - The byproduct of the optimization: complexity issues but not primary
>
>   1). **Complexity Analysis**: The essence of the SAM method lies in a more refined exploration of the gradient space, implicitly utilizing second-order information about the parameter space in the loss landscape. In experiments, it significantly improves generalization performance but incurs some unavoidable additional computations. The computational power consumption of the current algorithm complexity can be kept within a reasonable range and does not exponentially increase with the scale of data and models. This increase in complexity, compared to computational resources, is considered acceptable.
>
>   2). **Expansion and Solutions**: When scaling to more models or models with larger parameter sizes, SAM's gradient handling and parameter aggregation effects will be more beneficial than solely using co-teaching. The additional cost of gradient computations only linearly increases, which does not hinder the improvement in generalization performance. From a technical standpoint, considering the use of low-rank tricks and other existing technical tools can help reduce complexity, shifting complexity away from gradient computations as the core issue.

---

### Decision · Program_Chairs · 2024-09-25

**Decision:**

Accept (poster)

**Comment:**

This paper presents a probabilistic perspective on the co-teaching method, where the cleanliness of training data constitutes the hidden variables of the problem. Expectation Maximization (EM) can be naturally applied to this setting, but the authors argue that from an optimization viewpoint, EM may fail to produce high-quality optima for this problem. This motivates their adoption of the SAM optimizer instead of the EM algorithm to build their co-teaching algorithm. Their empirical results indeed confirm that utilizing SAM in co-teaching can consistently lead to better test accuracy across different levels and types of noise in the data.

The submission received 4 reviews, which raised some questions and concerns such as computational cost, asking for clarifications in the EM section, and how to extend the idea when co-teaching more than two networks. Authors provided a detailed response to these questions, as a result of which 3 reviewers increased and 1 reviewer decreased their initial rating. The final ratings are all in the accept zone, and overall assessment of reviewers indicates that they find the contributions of the paper non-trivial and interesting. In concordance with them, I recommend accept.